# Pain and Lethality Induced by Insect Stings: An Exploratory and Correlational Study

**DOI:** 10.3390/toxins11070427

**Published:** 2019-07-21

**Authors:** Justin O. Schmidt

**Affiliations:** Southwestern Biological Institute, 1961 W. Brichta Dr., Tucson, AZ 85745, USA; ponerine@dakotacom.net; Tel.: +1-520-884-9345

**Keywords:** venom, pain, ants, wasps, bees, Hymenoptera, envenomation, toxins, peptides, pharmacology

## Abstract

Pain is a natural bioassay for detecting and quantifying biological activities of venoms. The painfulness of stings delivered by ants, wasps, and bees can be easily measured in the field or lab using the stinging insect pain scale that rates the pain intensity from 1 to 4, with 1 being minor pain, and 4 being extreme, debilitating, excruciating pain. The painfulness of stings of 96 species of stinging insects and the lethalities of the venoms of 90 species was determined and utilized for pinpointing future directions for investigating venoms having pharmaceutically active principles that could benefit humanity. The findings suggest several under- or unexplored insect venoms worthy of future investigations, including: those that have exceedingly painful venoms, yet with extremely low lethality—tarantula hawk wasps (*Pepsis*) and velvet ants (Mutillidae); those that have extremely lethal venoms, yet induce very little pain—the ants, *Daceton* and *Tetraponera*; and those that have venomous stings and are both painful and lethal—the ants *Pogonomyrmex*, *Paraponera*, *Myrmecia*, *Neoponera*, and the social wasps *Synoeca*, *Agelaia*, and *Brachygastra*. Taken together, and separately, sting pain and venom lethality point to promising directions for mining of pharmaceutically active components derived from insect venoms.

## 1. Introduction

Stinging insects in the immense order Hymenoptera display a dazzling array of lifestyles and natural histories. These complex life histories offer a wealth of opportunities for the discovery of new natural products and pharmaceuticals to benefit the human endeavor. Each of the multitude of independent biological paths followed by stinging ants, social wasps, social bees, and solitary wasps and bees has resulted in evolutionary complex—and often unique—blends of venom constituents. Compared with the venoms of snakes, scorpions, medically important spiders, and a variety of other marine and terrestrial venomous animals, the venoms of most stinging insects are understudied. The reason for fewer investigations of insect venoms is explained, in part, by their general low potential for causing severe acute or long-term medical damage and partly by their small size. Additional complicating factors contributing to less emphasis on investigations of stinging insect venoms are the difficulties of identifying the insects and obtaining enough venom for study. Much of the recent research on insect venoms has focused on the relatively small number of species that are responsible for inducing human allergic reactions to insect stings [1]. The topic of sting allergy will not be addressed here.

Venoms of stinging insects have a variety of biologically important activities including the abilities to induce pain, cause cellular or organ toxicity, be lethal, produce paralysis, plus others [2,3,4,5]. These activities are the result of a wide variety of venom components, especially peptides and proteins, but also other categories of constituents [6,7,8]. The ability to cause pain is fundamental to most insect venoms that are used for defense against predators [9]. Pain is the body’s warning system that damage has occurred, is occurring, or is about to occur. In essence, pain informs the inflicted organism that it should immediately act to limit injury, or potential injury. The envenomated animal often releases the offending stinging insect and flees the area [9]. The net effect is that the stinging insect frequently survives the ordeal with minimal, or no, injury and for a social species enhances the survival of her nest mates (personal observation).

An understanding of the biology and use of the venom by a stinging insect species helps to guide strategies for discovery of new pain-inducing materials. Venoms used offensively for prey capture are predicted to produce little or no pain in the envenomated prey. The induction of pain in a prey animal would likely be detrimental to the predator by causing heightened flight, resistance, and potential for prey escape. Pain can also cause stress and increased physiological activity in the prey that, in turn, might reduce its survival time as a paralyzed food source for the young of the stinging insect. In a few species that use venom for paralyzing prey, the venom might also be used for defense. These venoms could contain pain-inducing constituents that would be predicted to be non-paralytic, but might be toxic or lethal to potential predators [9,10]. 

Pain sensation in humans is a subjective feeling ultimately registered by the brain. Consequently, quantitative and reliable assays for measuring conscious pain induced by individual venom components are scarce, though a variety of assays for measuring pain response in animals, including the rat paw lifting and/or licking assays have been developed [11,12]. Additionally, a variety of in vivo assays for nociception of pain by receptors, especially TRPV1 and other members of the transient receptor potential family of receptors, and the Nav channels, are known [13,14,15,16]. The shortage of simple metrics for measuring pain in humans has hampered our scientific ability to analyze the pain-causing properties of insect venom components. The result is that the evaluation of venom-induced human pain is often indirect and by inference. Investigators sometimes rely on personally testing the material on themselves, a procedure with inherent disadvantages and possible risks [17]. The limited number of human-based assays is partly responsible for the small number of characterized insect venom algogens reported in the literature. To help quantify painfulness of an insect sting, our group and colleagues developed the semi-quantitative stinging insect pain scale that rates the pain produced by an insect sting on a scale of 1 to 4 [18]. In the scale, 1 represents minor, almost trivial, pain and 4 represents the most extreme pain experienced. This insect sting pain rating can assist in choosing promising insect venoms for discovery of new algogens and medical products.

Two major components, phospholipases (A_1_ and/or A_2_) and hyaluronidases are nearly universally present in insect venoms. Additionally, several insect venoms contain esterases and lipases and sometimes acid phosphatases [19]. In addition to these major components, insect venoms contain a vast diversity of proteins and peptides in trace levels [6,8,20,21]. Known algogens in insect venoms include, among others, the peptide melittin from honeybee venom [16], wasp kinins in social wasp venoms [22], poneratoxin from the ant *Paraponera clavata* [23], peptide MIITX_1_-Mg1a from a bulldog ant [24], piperidine alkaloids in fire ants [25], barbatolysin in harvester ant venom [26], and possibly bombolitin in bumblebee venom [27]. Stings of virtually all social wasps, social bees, and ants cause at least some pain in humans. A few solitary wasp and bee species can also sting painfully. In most of the species of stinging insects, the properties of the pain-causing venom components are unknown. The intensity of the pain caused by an insect sting depends upon several factors, including the size of the stinging insect, the amount of venom it injects and, most importantly, on the chemical properties of the pain-inducing constituent(s). The purpose of this investigation was to explore as wide a diversity of stinging insects as possible to determine their ability to cause pain, and to pinpoint species that hold promise for discovering new pain-producing products that might be of benefit for science or medical investigations. The secondary purpose was to explore the lethality of venoms, again having in mind pinpointing potential species that hold promise for new scientific or medical discoveries. Several new species of stinging insects whose venoms hold promise are highlighted.

## 2. Results and Discussion

### 2.1. Pain Ratings of Insect Stings

Table 1 is a complete listing of all 115 stinging insects in 67 genera that were evaluated for the painfulness of their stings and/or the lethality of their venoms. Of these, sting pain determinations were made for 96 species in 62 genera including 38 ants in 27 genera, 25 social wasps in 12 genera, 6 social bees in 2 genera, 12 solitary bees in 10 genera, and 15 solitary wasps in 11 genera. The average pain level among the groups was: ants—1.62, social wasps—2.18, social bees—1.92, solitary bees—1.25, and solitary wasps—1.63. The members of each group do not necessarily represent the overall group in the natural world, instead represent those taxa that were often targeted for investigation, were historically known for painful stings, or were available. In many examples, the species were also among the largest in their respective genus, or the usual size of the individuals in the genus was large compared to their grouping in general. This was particularly true for the solitary bees and solitary wasps, most of which represent some of the largest known individuals in those categories. Given the targeted search for the most painful and lethal species of stinging insects, a general prediction is that most species not evaluated will deliver less painful stings than those represented in Table 1, or if they are in a genus that is listed in the table, their pain rating will be similar. The prediction of similarity of stings within a genus is based upon the sting pain values among the several species within the genera in Table 1. An extreme example of this similarity within a genus is found within the ant genus *Pogonomyrmex* in which all 21 species have the same rating of 3 on the sting pain scale. Similar results are found among the ant genera *Myrmecia* and *Solenopsis,* the social wasp genus *Vespula,* and the honeybee genus *Apis.*

### 2.2. Lethality of Stinging Insect Venoms 

The venom lethalities for 90 stinging insects in 50 genera are listed in Table 1. Lethalities were determined for 40 ants in 26 genera, 31 social wasps in 12 genera, 6 social bees in 2 genera, 8 solitary bees in 6 genera, and 5 solitary wasps in 4 genera. Overall, the venom lethalities of social bees and social wasps were higher than those of their solitary counterparts, with average values for the groupings: social wasps—5.38 mg/kg; social bees—5.75 mg/kg; solitary bees—37.1 mg/kg; and solitary wasps—77.6 mg/kg. The small number of solitary bees and solitary wasps is mainly because the venom of many individuals needed to be pooled for the lethality determinations. In addition, the general low overall lethality of solitary bees and wasps precluded extensive research on venom toxicity of these two groups. The ants presented a much higher variability in their venom lethalities: 13 taxa having lethalities of < 5 mg/kg, 10 in the range of 5–10 mg/kg, 8 in the range of 10–20 mg/kg, 7 in the range of 20–50 mg/kg, and 2 in the range of 50–128 mg/kg. This large range of values did not depend upon the body size of the ants: some ants of similar body weights had high lethalities (*Tetraponera* sp., 15 mg; *Pogonomyrmex* spp., 16 mg), medium lethalities (*Anochetus inermis*, 15 mg), or low lethalities (*Ectatomma ruidum*, 20 mg; *Harpegnathos venator*, 20 mg). The same contrast in lethalities was also observed among the largest ants, with the venoms of some species being highly lethal (*Myrmecia gulosa*, 80 mg; *Paraponera clavata*, 200 mg), some being moderately lethal (*Dinoponera gigantea*, 400 mg), and some being of low lethality (*Megaponera analis*, 90 mg) (weights, unpublished data).

### 2.3. Relationship between Sting Pain Level and Lethality of Stinging Insect Venoms

The question addressed here is the possible connection between the painfulness of a sting and the lethality of the venom delivered by the stinging insect. Of the 115 taxa investigated, data for both the sting pain rating and for the lethality are available for 71 stinging insects (Figure 1). The data are scattered throughout both the range of pain levels and lethalities with no apparent pattern or relationship, and no significant regression was found (r^2^ = 0.013; P = 0.356; line drawn only for visual reference). To obtain visual representations and possible relationships among the different stinging insect groups, the data were plotted separately for the ants, the social wasps, the social bees, and the solitary bees and wasps (Figure 2). The ant data scatter throughout Figure 2A and parallel the entire range found for all stinging species. Again, no relationship between sting painfulness and lethality was observed (r^2^ =0.028; P = 0.354). The social wasp data clump in the middle range of lethalities and rang from lowest to highest in painfulness (Figure 2B) with no relationship observed (r^2^ = 0.095; P = 0.162). The sting and venom activities of the six social bees exhibit midranges for both activities (Figure 2C). The data for social bees are limited in part because all stinging social bees reside in only two genera and the species within each genus have similar values. The solitary bees and wasps represent a wide variety of families and genera that have little in common biologically except for their solitary lifestyles. The 10 species share in common a comparatively low lethality, but have a maximal range from trivial to extreme in ability to deliver pain (Figure 2D). The relationship between pain and lethality among the solitary species, though not statistically significant, appears inverse, with those species delivering the most painful stings generally also trending towards having the least lethal venoms (r^2^ = 0.398; P = 0.0504).

### 2.4. Relationship between Sting Pain Level, Lethality, and Sociality of Stinging Insects

Field observations and the data presented here tend to indicate that stings of social insect are more painful than those of solitary species. On average, the sting pain level the social species in Table 1 is 1.85 ± 0.71 (S.D.; n = 69) compared to 1.46 ± 0.94 (S.D.; n = 27) for solitary species (P = 0.032, t-test). Some exceptions to this trend exist and will be the discussed in detail later.

The overall lethality of venoms of social species of stinging insects is higher than for solitary species. On average the lethality of social insects is 10.6 ± 17.3 mg/kg (S.D.; n = 77) compared to 52.7 ± 33.2 mg/kg (S.D.; n = 13) for solitary species, a highly significant difference (P = 0.0001, t-test).

Although both the painfulness of stings and the lethality of the venoms of social insects are greater than for solitary insects, the two factors combined do not result in a significant correlation between them and sociality. This might seem counterintuitive but the presence of sociality appears not simply based on venom lethality alone, but rather a combination of venom lethality and the amount venom delivered in a sting, in combination with the number of individuals available to deliver stings. When the amount of venom delivered per sting is considered, the result is a significant correlation between sting pain and venom potency (P < 0.001) [28]. Venom lethality also strongly correlates with the population in a colony, and with the overall weight of the individuals within a colony (P < 0.001), thus indicating that higher sociality evolved in concert with increased effectiveness of their venoms with more populated colonies [28]. These factors of venom quantity per insect and colony weight and number of individuals will not be discussed further here as they do not relate directly to questions of identification of venom peptides and proteins or their activities.

### 2.5. Natural History of Stinging Insects and How It Can Help Guide Discovery of Interesting Venom Peptides, Proteins, and Other Natural Pharmaceuticals

The functions and activities of the venoms of stinging insects evolved in concert with their natural history. If the natural history of a species is mainly based upon procuring prey for feeding their young, as occurs in most solitary wasps, then the primary activity of the venom would be expected to be paralysis, or sometimes death, of the prey. Most solitary species of wasps do not have serious predation pressure exerted by large predators, especially vertebrate predators, and hence their stings and venoms are only rarely used for defense. Their stings and tend not to be highly painful or toxic to vertebrates. In contrast, social wasps never use their stings and venom for subduing prey. Powerful mandibles are used for prey capture and dismemberment and the sting is used only for defense (and in some situations for release of pheromones or other activities) [29]. Thus, in general, solitary and social wasps would be expected to have different venom chemistries and activities.

All ants are social. Ants also have an extreme breath of behaviors and natural histories. Some ants use their stings to paralyze or subdue prey, whereas others rarely use their venom for prey capture. All stinging ants use their stings and venom for defense against potential predators, whether the predators are small arthropods or large vertebrates. These diverse natural histories of ants provide a wealth of potential opportunities for discovery of new and exciting peptides, proteins, and other active constituents.

All bees are vegetarians, with the exception of a few species that scavenge dead animals. Bees, therefore, have no need to use their venom for prey capture and their stings and venoms are only used for defense against predators. In the case of solitary bees, their main predators are also small animals, mainly spiders, other arachnids, and insects, especially ants. Solitary bees rarely experience strong predation pressure vertebrates and their stings and venoms have not evolved to be especially painful or toxic to vertebrates. Social bees, mainly honeybees and bumblebees, live in colonies rich in resources including honey, pollen, and larvae and pupae that provide an enticing nutritional reward for mammals and birds. In response to this heightened predation pressure experience by social bees, their venoms have evolved to be lethal to vertebrates and to induce pain. 

### 2.6. Targeting Promising Species of Stinging Insects for Discovery of New Pharmaceuticals Based upon Sting Pain and Lethality

Species of stinging insects that exhibit extreme values of either sting painfulness or lethality may be promising for further investigation. Especially interesting might be those species whose stings are extraordinarily painful but have little lethal activity, or species that are the opposite with extremely lethal venoms that are not particularly painful. A third category of species that might be of interest are those that have both painful stings and are highly lethal. Species in a fourth category that have stings of low painfulness and their venoms are of low lethality likely have minimal potential for discovery of new interesting peptides or pharmaceuticals that relate to human biology or welfare. However, those species that are low in both categories might have high potential for discovery of peptides or other active principles that target insects and other invertebrates and could be of benefit for agriculture. Species in this category include many of the solitary wasps, with noteworthy species being the cicada killer wasps in the genus *Sphecius*, the potter wasps in the subfamily Eumeninae, and any of the species that routinely paralyze or kill insect or spider prey. The sting painfulness and/or venom lethality of many of these wasps is *Terra incognita* and is well worth investigating. Solitary wasp species such as velvet ants in the family Mutillidae that use their stings only for defense are likely to show no potential for discovering new agricultural materials. The main disadvantage of investigating solitary hunting wasps is the problem of obtaining enough individuals for study. 

Solitary bees use their stings and venom strictly for defense, and even for defense most of them have ineffective venoms that produce little pain and low toxicity. The activity of their venoms towards insects is basically unknown. Thus, solitary bees likely represent a group that have little or no potential for discovery of new peptides or materials useful for either agriculture or other human endeavors. The one exception to this generalization might be the large carpenter bees in the genus *Xylocopa* that have venoms that produce moderate pain and moderate lethality. An additional benefit of these bees is that they are large and easy to obtain.

Species whose stings produce extreme pain, yet have low venom lethality provide a promising starting point for the development of bioassays for screening of potential analgesic pharmaceuticals. They have active components that can readily induce pain, while not causing tissue toxicity. Tarantula hawks in the genus *Pepsis* and the velvet ants the family Mutillidae, both of which produce extraordinary painful stings, yet have almost no vertebrate lethality, are candidates for further study. One species that produces the most painful stings of any hymenopteran is the bullet ant *Paraponera clavata*. The venom of this species is also highly lethal and both activities appear to be caused mainly by the single peptide poneratoxin that has been well studied [23]. A promising pain-inducing venom that has potential for new meaningful discoveries is that of the warrior wasps in the genus *Synoeca.* The stings of these wasps are intensely painful for at least an hour and have a sting pain rating of 4. The venom is also highly lethal. This small genus of six species is widespread and common throughout much of tropical Latin America and their venoms have been studied to a limited extent [30]. These large wasps live in populist colonies and produce 270 µg venom/wasp (Schmidt, unpublished). A final promising group of wasps with painful stings and lethal venoms worthy of investigation of the fire wasps in the genus *Agelaia.* The genus of about two dozen species of small wasps live in populist colonies of many thousands of individuals and range throughout much of the New World tropics.

Stings that are highly lethal, yet induce low pain levels are relatively uncommon. Most impressive example of this is an unidentified species in the ant genus *Tetraponera* from Malaysia. The stings of this species produce only the mild pain level of 1, yet have the exquisite lethality of 0.35 mg/kg. This venom could be useful for assays designed to determine the mechanism of lethality while producing little pain. Another species of similar potential is that of the trap-jaw ant *Daceton armigerum* that is a common arboreal species in the tree canopy of the rain forests of northern South America. Other promising species include the common Latin American ant *Ectatomma tuberculatum* and many of the Australian bulldog ants in the genus *Myrmecia* that have been subject to a variety of studies [24,31]. The venoms of the social wasps in the enormous Old-World genus *Ropalidia* have been neglected and appear to have potential for new discoveries.

The final category of stinging insects that have promising venoms are those that are both painful and lethal. In addition to the already mentioned bullet ants, the new world harvester ants in the genus *Pogonomyrmex* present ideal opportunities. Their venoms are the most toxic known from any of the Hymenoptera and produce intense waves of deep, agonizing pain that lasts 4–8 h, plus induce piloerection and localized sweating at sting site [29,32]. These ants are abundant over large areas of North and South America and are easy to maintain in the laboratory. The Neotropical ants in the genus *Neoponera*, including *N. villosa* and the termite-hunting ant, *N. commutata* are large species whose stings are painful and venom is lethal to mammals and paralytic to insects [33,34]. The venoms of honey wasps in the Neotropical genus *Brachygastra,* have not been studied and represent a good opportunity for discovery of interesting venom activities and constituents. Likewise, many of the paper wasps in the large worldwide genus *Polistes* possess both painful stings and lethal venoms and their venoms are worthy of further investigation.

## 3. Materials and Methods 

### 3.1. Insects

Stinging ants, wasp, and bees were live collected from their natural environments, typically from their nests in the soil, in trees, sometimes in urban areas, or from their normal foraging locations on flowers, vegetation, or soil surface. Species were determined with keys to the various taxa, with difficult identifications made by experts in the specific taxa. Once collected, the insects were cooled on ice, and in most situations the iced insects were brought to the laboratory where they were frozen and stored at −20 °C until their use. In some field situations where access to a freezer was unavailable, the insects were maintained on ice until dissected for venom. In situations where the insects were maintained on ice, the insect tissues were fresh, appeared the same as live dissected individuals, and the venom reservoirs were intact and contained venom that was clear and transparent.

### 3.2. Pain Measurement

Since human perception of sting pain cannot be easily measured instrumentally or with great precision, a pain scale for the immediate, acute pain caused by a sting was developed [18,35]. The scale ranges in values from 1–4 and is anchored by the value of a single honeybee sting (*Apis mellifera*), which is defined as a 2 on the scale. The honeybee is a convenient reference point because honeybees exist worldwide, are abundant, and most people have been stung by a honeybee. They are also about midway within the range of pain intensities produced by hymenopterous stings. Sting pain induced by a single sting can vary depending upon how much venom was delivered with the sting, where on the body the sting occurred (for example stings to the nose, lips, or palms of hands are considerably more painful than stings to lower legs or arms; see also [19]), the age of the insect, the time of day the sting was received, and other factors [36]. For these reasons, the scale was limited to 4 values, plus a trivial value of 0 for insects that are incapable of penetrating human skin. The criteria distinguishing between pain levels is that the pain of the lower level is substantially less than the pain in the upper level and that the evaluating person would clearly know that one of the stings hurt considerably more than the other. When comparing species, the evaluator compares the current sting pain with memory of the pain of previous stings by a honeybee or other species for which the pain was rated previously. In most cases the reference point for the value of 2 that is used to stabilize the scale is quite robust because the evaluator has been stung many times by honeybees and can sense the amount of pain in an average honeybee sting. The number of stings for other species evaluated can vary from a low of a single sting to a high of many stings depending upon the species; consequently, some values have greater potential subjectivity than others. In some cases, values halfway between whole numbers are assigned where the pain appears greater than the lower level, yet less than the higher level. This evaluation system works remarkably well as witnessed by nearly identical ratings for stings by various colleagues (personal observations) [35]. Stings of many different species have the same numerical value; this does not imply that they are identical in feeling, but that they fall into the same general range of acute painfulness, and presumed effectiveness as predation deterrents. Pain that arises at or near the sting site hours or days after the initial sting pain has receded is not considered for this pain scale because it is caused by immunological or physiological reactions to the venom or its damage [37].

Most measurements of pain were scored in the field from live stings as they naturally occurred. In exceptional situations where normal stings were not received during the course of working with or collecting the species, or when the species does not normally sting as a primary defense, intentional stings were received by forcing the insect to sting the medial side of the forearm. This area was chosen because the low hair density allows better observation and that area is a convenient and relatively non-specialized part of the skin.

### 3.3. Venom

Pure venom was obtained by the method of Schmidt [36]. In brief, frozen ants or bees were thawed, their sting apparatuses removed to a spot of distilled water, the venom reservoir (minus filamentous glands) was pinched off at the duct and removed from the rest of the sting apparatus, twice rinsed with distilled water, and placed in clean distilled water (Figure 3). Depending upon the number of insects available and their size, up to 100 reservoirs were collected into an approximately 50 µL droplet of distilled water, after which the venom was squeezed from the reservoirs and the empty chitinous reservoirs were discarded. The pure venom was either lyophilized and stored at −20 °C until used, or dried over molecular sieves 5A (Supelco, Bellefonte, PA, USA) and then stored in a freezer −20 °C until used.

For wasps, venom was collected by expression through the sting shaft into the space (by capillary action) between the tines of fine forceps. Often, in order to accomplish this, one or two terminal sternites of the abdomen needed to be removed to allow the sting apparatus including the muscular venom reservoir to be removed. To facilitate venom expression gentle squeezing pressure was applied via broad forceps to the venom reservoir. After the venom was collected, it was released into the bottom of a small polyethylene microtube by opening the forceps and allowing the venom to be deposited in the microtube. The venom from several individuals could be combined a single microtube before the venom was frozen, lyophilized and stored at −20 °C.

### 3.4. Physiological Measurements

Animal experiments were approved by the Southwestern Biological Institute Ethics Committee (SWBIEC/0017_29, 29 June 2016). Dried venom was used in all tests. Venoms were weighed to the nearest 1.0 μg on a 7-place microbalance (Model 1-912, Mettler, Zurich, Switzerland). Damage potential of an insect sting was measured as lethality of the venom to ICR white mice of mixed sex and ranging in weight from 18 to 22 g, as described previously [38]. Generally, four groups of six mice were used for each experiment, though in some situations where venom availability was limited only three groups of four mice were used.

For each analysis the calculated and measured weight of venom was dissolved in 0.15 M NaCl saline and injected in the volume of 0.6% of the animal body weight. Venoms were intravenously (i.v.) injected into the tail veins of the mice. Median lethal amount of venom to kill 50% of the individuals (LD_50_) in 24 h were calculated according to the rapid 50% endpoint method that interpolates between the 25% and 75% values to obtain a reliable 50% lethality value [39].

## Figures and Tables

**Figure 1 toxins-11-00427-f001:**
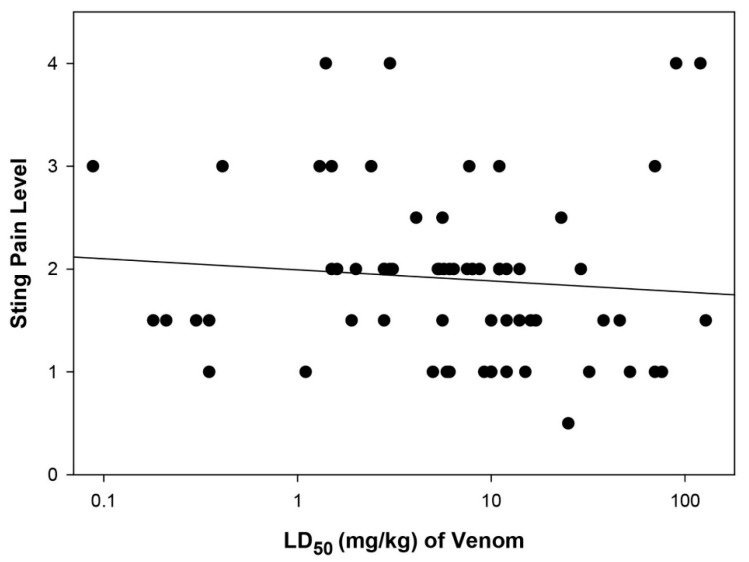
Scatter diagram of sting pain level and lethality of all 71 species of Hymenoptera for which both values are available. The trendline is provided only for reference, as no significant trend was observed (r^2^ = 0.013; P = 0.356).

**Figure 2 toxins-11-00427-f002:**
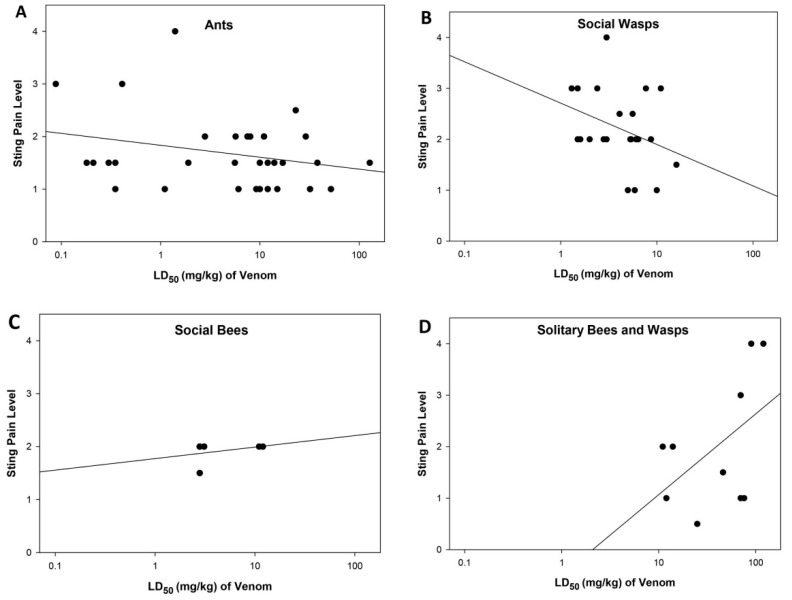
Scatter diagrams of sting pain level and lethality showing potential trends among the taxa within the individual groupings of stinging Hymenoptera. (**A**) The 33 species of ants, (**B**) the 22 species of social wasps, (**C**) the 6 species of social bees, and (**D**) the 10 species of solitary bees and wasps. The ants exhibit the broadest range of values, while the values of the other groupings are more tightly clustered. The trendlines are provided only for reference, as no significant relationship between sting pain level and lethality was observed for any of the groups ((**A**) r^2^ = 0.028. P = 0.354; (**B**) r^2^ = 0.095, P = 0.162; (**C**) r^2^ = 0.105, P = 0.532; (**D**) r^2^ = 0.398, P = 0.0504).

**Figure 3 toxins-11-00427-f003:**
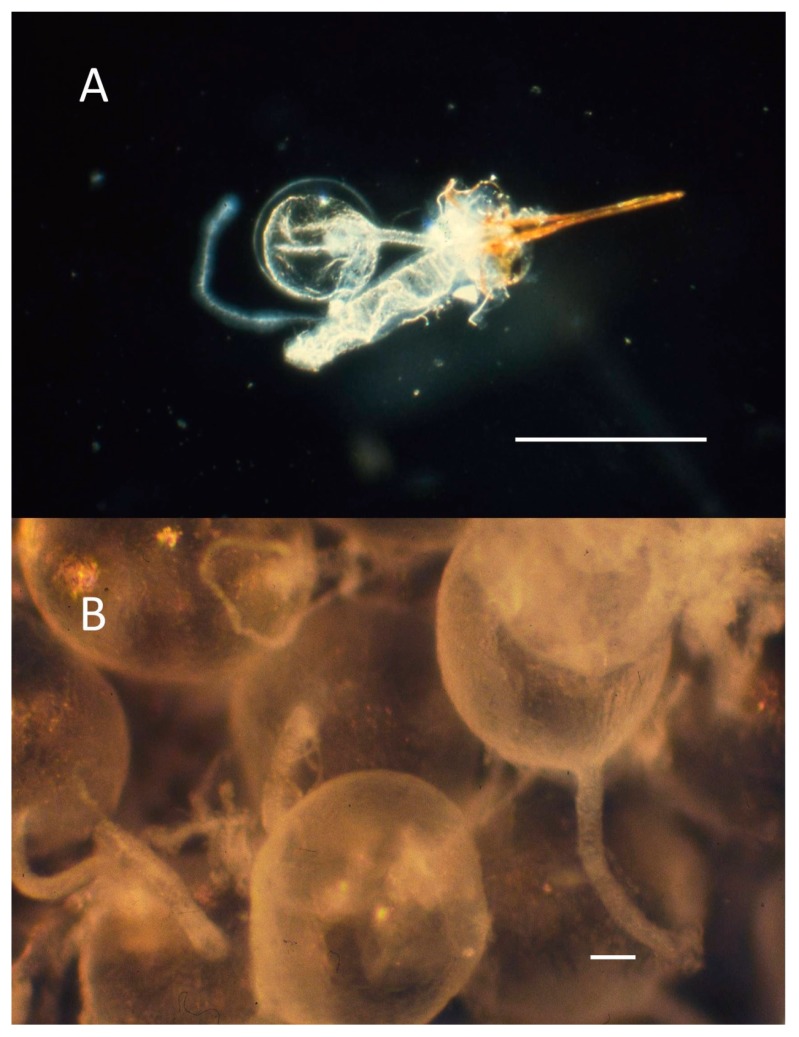
(**A**) Sting apparatus of *Pogonomyrmex badius* showing sting shaft, tubular Dufour’s gland, and spherical venom reservoir with a long venom duct leading to the base of the sting shaft (scale bar = 1 mm). (**B**) Isolated venom reservoirs of *Pogonomyrmex maricopa* in a droplet of distilled water ready for the venom to be drained and the empty membranous reservoirs discarded (scale bar = 0.1 mm). Photos taken with an Olympus PM-10-A camera attached to an Olympus JM Zoom Stereo Microscope.

**Table 1 toxins-11-00427-t001:** Sting pain rating on a scale of 1 to 4 and venom lethality of ant, social wasp, social bee, and solitary species of stinging Hymenoptera. The data are arranged by increasing pain level from the lowest rated species in each genus, followed by those genera unrated for pain and, within a pain level, arranged by highest to lowest lethality. Blanks in the table columns indicate no data are available for the assay.

Species (Common Name)	Sting Pain	LD_50_ (mg/kg)
**Ants**		
*Solenopsis invicta* (red fire ant)	1	
*S. xyloni* (southern fire ant)	1	
*S. geminata* (tropical fire ant)	1	
*Tetraponera* sp. (Old World twig ant)	1	0.35
*Daceton armigerum* (trap-jawed ant)	1	1.1
*Myrmica rubra* (European fire ant)	1	6.1
*Bothroponera strigulosa*	1	9.2
*Leptogenys kitteli*	1	10
*Pseudomyrmex gracilis* (twig ant)	1	12
*P. nigrocinctus* (bullhorn acacia ant)	1.5	1.9
*Ectatomma ruidum,*	1	15
*E. tuberculatum*	1.5	0.3
*E. quadridens*	1.5	17
*Ectatomma* sp.		17
*Opthalmopone berthoudi* (big-eye ant)	1	32
*Harpegnathos venator*	1	52
*Brachyponera chinensis* (needle ant)	1	
*B. sennaarensis* (Samsum ant)	1.5	5.6
*Myrmecia gulosa* (red bulldog ant)	1.5	0.18
*M. browning* (bulldog ant)		0.18
*M. tarsata* (bulldog ant)		0.18
*M. simillima* (bulldog ant)	1.5	0.21
*M. rufinodis* (bulldog ant)	1.5	0.35
*M. pilosula* (Jack jumper ant)	2	5.7
*Eciton burchelli* (army ant)	1.5	10
*Anochetus inermis* (a trap-jaw ant)	1.5	12
*Dinoponera gigantea* (giant ant)	1.5	14
*Paltothyreus tarsatus* (giant stink ant)	1.5	38
*Megaponera analis* (Matabele ant)	1.5	128
*Pachycondyla crassinoda*	2	2.8
*Neoponera villosa*	2	7.5
*N. commutate* (termite-hunting ant)	2	11
*Streblognathus aethiopicus* (African giant ant)	2	8
*Diacamma rugosum*	2	8
*Platythyrea lamellose*	2	11
*P. cribrinodis*		42
*Odontoponera transversa*	2	29
*Rhytidoponera metallica*	2	
*Odontomachus bauri* (trap-jaw ant)	2.5	23
*O. infandus* (trap-jaw ant)		33
*O. chelifer* (trap-jaw ant)		37
*Pogonomyrmex cunicularius* (Argentine harvester ant)	3	0.088
*Pogonomyrmex* (North American harvester ants) (20 spp.)	3	0.12–0.7
*Paraponera clavata* (bullet ant)	4	1.4
*Manica bradleyi*		6
**Social Wasps**		
*Polybia occidentalis* (polybia wasp)	1	5
*P. rejecta* (polybia wasp)	1.5	16
*P. simillima* (polybia wasp)	2.5	4.1
*P. sericea* (polybia wasp)		6.1
*Ropalidia flavobrunnea*	1	5.9
*Ropalidia* sp.	1	10
*Ropalidia (Icarielia)* sp.		14
*Belonogaster* sp. (thin paper wasp)	1.5	
*B. juncea colonialis* (fire-tail wasp)	2	3
*Brachygastra mellifica* (honey wasp)	2	1.5
*Vespula germanica* (yellowjacket wasp)	2	2.8
*V. vulgaris* (yellowjacket wasp)	2	5.4
*V. pensylvanica* (yellowjacket wasp)	2	6.4
*V. vidua* (yellowjacket wasp)		2.6
*V. consobrina* (yellowjacket wasp)		2.8
*Polistes instabilis* (paper wasp)	2	1.6
*P. arizonicus* (paper wasp)	2	2
*P. infuscatus* (paper wasp)	3	1.3
*P. erythrocephalus* (paper wasp)	3	1.5
*P. canadensis*. (paper wasp)	3	2.4
*P. tepidus* (paper wasp)	3	7.7
*P. annularis* (paper wasp)	3	11
*Parachartergus fraternus* (artistic wasp)	2	5.3
*Dolichovespula maculata* (baldfaced hornet)	2	6.1
*D. arenaria* (aerial yellowjacket)	2	8.7
*Mischocyttarus* sp. (a paper wasp)	2	
*Agelaia myrmecophila* (fire wasp)	2.5	5.6
*Provespa* sp. (nocturnal hornet)	2.5	
*Synoeca septentrionalis* (warrior wasp)	4	3
*Vespa luctuosa* (hornet)		1.6
*V. tropica* (hornet)		2.8
*V. simillima* (hornet)		3.1
*V. mandarinia* (giant hornet)		4.1
*Apoica pallens* (night wasp)		13.5
**Social Bees**		
*Apis florea* (dwarf honey bee)	1.5	2.8
*A. mellifera* (honey bee)	2	2.8
*A. dorsata* (giant honey bee)	2	2.8
*A. cerana* (Eastern honey bee)	2	3.1
*Bombus impatiens* (bumble bee)	2	11
*B. sonorus* (bumble bee)	2	12
**Solitary Bees**		
*Dieunomia heteropoda* (giant sweat bee)	0.5	25
*Triepeolus* sp. (cuckoo bee)	0.5	
*Xenoglossa angustior* (squash bee)	1	12
*Habropoda pallida* (white-faced bee)	1	70
*Diadasia rinconis* (cactus bee)	1	76
*Emphoropsis pallida*	1	
*Lasioglossum* spp. (sweat bee)	1	
*Ericrocis lata* (cuckoo bee)	1	
*Euglossa dilemma* (orchid bee)	1.5	
*Xylocopa rufa* (nocturnal carpenter bee)	2	11
*X. californica* (carpenter bee)	2	14
*X. veripuncta* (carpenter bee)		33
*Xylocopa* sp. (giant Bornean bee)	2.5	
*Centris pallida* (palo verde bee)		56
**Solitary Wasps**		
*Sapyga pumila* (club-horned wasp)	0.5	
Eumeninae spp. (potter wasps)	1	
*Sphecius convallis* (cicada killer wasp)	1	
*S. grandis* (cicada killer wasp)	1.5	46
*Sphex pensylvanicus* (great black wasp)	1	
*Chlorion cyaneum* (cockroach-hunter wasp)	1	
*Triscolia ardens* (scarab-hunter wasp)	1	
*Sceliphron caementarium* (mud dauber wasp)	1	
*Euodynerus crypticus* (water walking wasp)	1	
*Dasymutilla thetis* (little velvet ant)	1	
*D. gloriosa* (velvet ant)	2	
*D. klugii* (cow killer velvet ant)	3	70
*Pepsis grossa* (tarantula hawk wasp)	4	90
*P. thisbe* (tarantula hawk wasp)	4	120
Mutillidae sp. (small nocturnal velvet ant)	1.5	
*Crioscolia flammicoma* (scoliid wasp)		62

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
