# Peer review of "Pain and Lethality Induced by Insect Stings: An Exploratory and Correlational Study"

_toxins, 2019, doi:10.3390/toxins11070427_

Round 1

Reviewer 1 Report

The auhtors have modified properly the manuscript, making it minimally acceptable for publication.

Reviewer 2 Report

I'm glad to see that methods section was improved, which was the major problem of original submission. In general, the issues that were pointed out were fixed and now I believe the manuscript is suitable for publication. As a last recommend, I would only  improve the table 1 legend and include that LD50 was obtained in ICR white mice.

Reviewer 3 Report

Good reply! Congratulations!

This manuscript is a resubmission of an earlier submission. The following is a list of the peer review reports and author responses from that submission.

Round 1

Reviewer 1 Report

The present manuscript aims to stablish a scale of pain for the crude venom of a series of venomous Hymenoptera insects, and also try to correlate to intensity of pain with the lethality of each venom.  The present manuscript has serious concerns as reported bellow:

The manuscript as whole is very speculative, since the main results (pain intensity measurement) were acquired just by using human individuals as subjects of study; it is important to emphasize that there are series of different experimental protocols to evaluate pain in the literature. None of these methods were used in the present study. The use of a human feeling of pain is very subjective; not even can be easily reproduced when comparing different subject individual. Different individuals may feel different intensities of pain for the same amount of inoculated venom. In the present study not even the amount of venom used in each observation was standardized.

There different biochemical and pharmacological process involved in pain occurrence; thus, depending on the composition of each venom, different types and level of intensity of pain may be observed (or not). This important aspect was not considered in the present study.

If the authors had used any technic protocol already consolidated in the literature for evaluating pain intensity, the manuscript could be more seriously considered. If the main objective of to measure pain intensity, to try correlate this value (pain intensity) with LD50 and eusociality of the Hymenoptera, all the correlations were seriously affected by the subjectivity of this evaluation.

The methodology of determination of LD50 was poorly described, contributing to make very difficult to evaluate the significance of this type of result, and the possibilities of correlations with pain intensity.

The manuscript is not recommended for publication in the present format.

Reviewer 2 Report

The manuscript entitled  “The Insect Sting Pain Scale: How the Pain and Lethality of Ant, Wasp, and Bee Venoms Can Guide the Way for Human Benefit” aims to characterize the pain of many insects and correlated it with the medium lethal dose as a tool for discovery of new bioactive molecules. The idea of the manuscript is interesting, however, I have several concerns. Some of the basic information were also not provided or are not clearly present. In my opinion, the manuscript in the present format is not suitable for acceptance and it should be thoroughly revised before being acceptance. The following points may be helpful for the authors to improve the quality of this manuscript.

In general, I miss more references during the text. For example, during the introduction, paragraph from line 49 to line 56 has no references, which continues on Line 57 until line 64 that discuss about the lack of methods to measure pain without subjectivity. Would be nice to increase the number of references.

I also found very confusing the association with the results with the discovery of new bioactive molecules.

Results and discussion

Line 120-121 “This large range of values did not depend upon the body size of the ants”

Would be nice to provide the regular body size of each ant, of references about it.

Is it not possible to compare LD50/ pain results with other works?

Figure 2 - Although it shows no statistical relevance, it would be nice to add into the figure the correlation and p value.

Line 163-164  "On average the lethality of social insects is 10.6 ± 17.3 mg/kg (S.D.; n = 77) compared to 52.7  ± 33.2 mg/kg (S.D.; n = 13) for solitary species, a highly significant difference (P = 0.0001, t-test)."

The values of SD is very high (more than 60%), even for an in vivo assay. How could you explain that? How many replicates were made?

Methods

-  In general, methods section is very confusing.

1)  Line 270 “Stinging ants, wasp, and bees were live collected from the field”

What field? It is very important to provide correct location and include the geo-localization of the place. Moreover, how many animals were collected? Were the animals from same species collected from the same colony, or more colonies? And how the animals were taxonomically identified?

2)  Line 272-273 "In some field situations where access to a freezer was unavailable, the insects were maintained on ice until dissected for venom"

What animals? Could this cause any interference?

3) I miss the information about the model that was used. The model should be described on results and on table 1. The authors only refer about the  ”model animal, the mouse”, at line 332. But what kind of mouse (Balb? Black? Nude? Other?) were used to determine pain and LD50? What age and weight did the mouse had? How the experiment was conduct? How many mouse were used for the LD50 experiment?

4) About the venom used – Why did you weight only 1 ug? Was the microbalance appropriated calibrated to assure this measurement? Can you estimate the ratio of protein per dry weight? Did you try to quantify using any biochemical method?

Reviewer 3 Report

See document attached. 

Reviewer 4 Report

Very interesting research! Authors studied the venom of stinging insects from 115 species. The report will be proved important for future investigation of cell membrane receptors and understanding the toxicity of insects venom. I have two suggestions:

1, Please provide the detailed methods that used for measuring the pain. What kind of mouse was used? How many repeats for one measuring? What is your control? etc.

2, It would be great if authors can introduce the major components of insect venom and the major pain pathways in mouse or human.